**Data Availability Statement:** The data is available at: https://luna16.grand-challenge.org/.

**Funding:** This work is partially supported by Zhejiang Province Philosophy and Social Science Planning Project under Grant No. 23NDJC165YB;

# Effective and efficient content-based similarity retrieval of large lung CT images based on WSSLN model

**Yi Zhuang**[1]*, **Nan Jiang**[2]

**1** School of Computer Science & Technology, Zhejiang Gongshang University, Hangzhou, P.R. China,
**2** Affiliated Hangzhou First People's Hospital, Zhejiang University School of Medicine, Hangzhou, P.R. China

* zhuang@zjgsu.edu.cn

## Abstract

The in-depth combination and application of AI technology and medical imaging, especially high- definition CT imaging technology, make accurate diagnosis and treatment possible. Retrieving similar CT image($CI$)s to an input one from the large-scale $CI$ database of labeled diseases is helpful to realize a precise computer-aided diagnosis. In this paper, we take lung $CI$ as an example and propose progressive content-based similarity retrieval(CBSR) method of the lung $CI$s based on a _Weakly Supervised Similarity Learning Network_ (WSSLN) model. Two enabling techniques (i.e., _the WSSLN model_ and _the distance- based pruning scheme_) are proposed to facilitate the CBSR processing of the large lung $CI$s. The main result of our paper is that, our approach is about 45% more effective than the state-of-the-art methods in terms of the mean average precision($mAP$). Moreover, for the retrieval efficiency, the _WSSLN_-based CBSR method is about 150% more efficient than the sequential scan.

## 1. Introduction

With the in-depth fusion of AI technology and medical imaging technology, content-based high- definition medical image retrieval(CBMIR), especially for content-based CT image($CI$) retrieval, plays an increasingly important role in the field of computer-aided disease diagnosis [1]. In most cases, the $CI$s include not only information about the images themselves but also a list of patient-specific therapeutic methods. Through $CI$ comparisons, physicians can identify similar $CI$s from various individuals who have a high risk of contracting a same disease because they include the same clinical symptoms by comparing their $CI$s. So retrieving similar $CI$s to an input one from the large-scale $CI$ database of labeled diseases is helpful to realize computer-aided diagnosis. The paper takes lung $CI$ as an example.

Compared with the similarity retrieval of ordinary images, the medical image (e.g., $CI$) similarity retrieval requires higher retrieval accuracy. Although all $CI$s contain blood vessels, bones and soft tissues (e.g. thorax, trachea, bronchi, etc.) inside the lung lobes and the lung $CI$s are generally similar from person to person, the shape of the lung lobes and details about the bronchi, blood vessels, and nodules inside the lungs vary from patient to patient. At the pixel

Zhejiang Provincial Natural Science Foundation of China under Grant No. LY22F020010; the Zhejiang Public Welfare Technology Application Research Project under grant No. LGF22H180039, LTGY23F020002; the Zhejiang Traditional Chinese Medicine Science and Technology Project under grant No. 2023ZL119. There was no additional external funding received for this study.

**Competing interests:** The authors have declared that no competing interests exist.

level, they are all different. As a result, high precision similarity retrieval of the lung $CI$s is generally required. In addition, the objects inside the lung lobes have complex characteristics such as location and shape. So it is difficult to describe and quantify them. Meanwhile, deep learning-based similarity assessment usually requires a large amount of data and labels. The manually labeling of a great amount of $CI$s by the medical experts is a time-consuming, laborious and expensive task. Content-based similarity retrieval (CBSR) of the $CI$s usually combines feature descriptors extracted by deep learning networks with traditional mathematical methods. Their similarity is then calculated based on a distance formula. This retrieval method, however, requires a large number of labels to annotate the $CI$s which consumes a large amount of storage space.

To address the above challenges, the paper introduces a *Weakly Supervised Similarity Learning Network* (WSSLN) model for progressive content-based similarity retrieval of the lung $CI$s. As a new deep learning network design, the *WSSLN* model has two layers: 1) *the first layer of the network is in charge of calculating the similarity of contour information and incorporates a spatial transformation layer (STL) prior to training*; 2) *the second layer of the network is in charge of calculating the similarity of the details within the lung lobe*. To reduce the high computation costs of the similarity comparison via the *WSSLN* model, a distance-based pruning strategy is developed that may effectively reduce the search space during the *WSSLN*-based CBSR processing.

The main contributions of this paper are as follows:

1. We propose a progressive content-based similarity retrieval method of the large lung $CI$ images based on the *WSSLN* model.

2. We present a new weakly supervised deep learning network called the *WSSLN*, for lung $CI$ similarity assessment, in which a new automatic labeling method for similarity labeling between $CI$s is developed.

3. We propose a distance-based pruning scheme to effectively reduce the search space.

4. Extensive experiments are conducted to demonstrate the effectiveness and efficiency of our proposed *WSSLN*-based CBSR method.

## 2. Background

### 2.1 Content-based medical image retrieval

The CBMIR system involves the process of feature extraction, similarity measurement and ranking of medical images. The key lies in the feature extraction of a medical image that has gone through two stages.

The first stage is visual feature extraction which consists of global feature and local feature extraction [2]. Mizotin et al. [3] present a brain magnetic resonance image (MRI) retrieval method based on SIFT features of visual bag-of-words (BoVWs) for the diagnosis of Alzheimer's disease. The Idiap research team [4] coupled LBP and modSIFT [5]. The two descriptors obtained an error of 178.93 on the IRMA dataset. Pan et al. [6] use an *Uncertain Location Graph* (ULG) structure to model the brain $CI$s by which the accuracy is up to 80%. Karthik et al. [7] propose a hybrid feature model to represent a hybrid feature vector of shape and texture properties. Sampathila et al. [8] design a CBIR method using image features (e.g., color, shape, and texture) to represent and retrieve similar images in a large database.

The second stage is semantic feature extraction which can lead to more accurate retrieval results. Shin et al. [9] apply the migration learning ideas to fine-tune CNN models pre-trained

on ImageNet datasets for the interstitial lung disease dataset (ILD) and thoracoabdominal lymph node(LN) dataset. Sundararajan et al. [10] propose a method to retrieve avascular necrosis-free(AN) images using deep bilinear convolutional network (DB-CNN) feature representation. Khatami et al. [11] first predict the most probable class of the retrieved image by CNN network, and then applied it in the search space consisting of this class Radon transform for further retrieval. After that, Khatami et al. [12] propose two more retrieval schemes. Ma et al. [13] try to fuse the semantic and visual similarities between the two images as their similarity.

In addition, Lai et al. [14] present a deep neural network hashing (DNNH) method that describes more complex semantic information by using triplet-based constraints. Liu et al. [15] propose a deep supervised hashing method (DSH) to support fast image retrieval. Anwar et al. [16] present a novel image retrieval method based on a combination of local and global histograms of visual words. Mehmood et al. [17] design an image retrieval scheme based on rectangular spatial histograms of visual words. Mehmood et al [18] propose a content-based image retrieval and semantic annotation method based on the weighted average of triangular histograms using support vector machine. Cai et al. [19] design a new loss function based on CNN with hash coding to learn models to make images belonging to the same class with similar features, and the proposed method was TCIA-CT dataset achieved satisfactory results. Bibi et al. [20] put forward a multimodal framework for content-based image retrieval. To boost the performance of the BoVW model, Baig et al. [21] adopt the SURF–CoHOG-based sparse features with relevance feedback for CBIR.

## 2.2 The AP cluster algorithm

The *affinity-propagation*(AP)-based clustering method [27] is an iterative algorithm in which each data point can be viewed as a network node which passes messages to other nodes in order to determine which nodes should be exemplars and which nodes should be associated with those exemplars. An exemplar is the point which best represents other points in its cluster.

To maximize the overall similarity of all data points to their exemplars, the algorithm is based on the ideas of belief-propagation. There are two types of messages sent between data point *i* and candidate exemplar *k*: *responsibility r(i,k)* and *availability a(i,k))*. Responsibility messages are sent from *i* to *k* and reflect how strongly data point *i* favors *k* over other candidate exemplars. Availability messages are sent from *k* to *i* and reflect how available *i* is to be assigned to *k*.

The messages are passed during several iterations in which the evidence accumulates that some points are better exemplars. The algorithm reaches convergence when enough evidence has been formed about exemplars and assignments to exemplars. At this stage node *i* is assigned to whichever candidate exemplar *k* maximizes the value of $a(i,k)+r(i,k)$. If this value is maximized where $i = k$ then *i* itself is an exemplar. $a(i,k)$ is initialized to a zero value so that $r(i,k)$ can be calculated in the first iteration. After this the availabilities are calculated and stored to be ready for the next iteration.

## 3. Methodologies

### 3.1 System framework

In this section, we provide a system framework for the retrieval system in which three stages are depicted in Fig 1: 1) *data generation and training stage* (offline), 2) *preprocessing stage* (offline), and 3) *CI retrieval stage* (online).

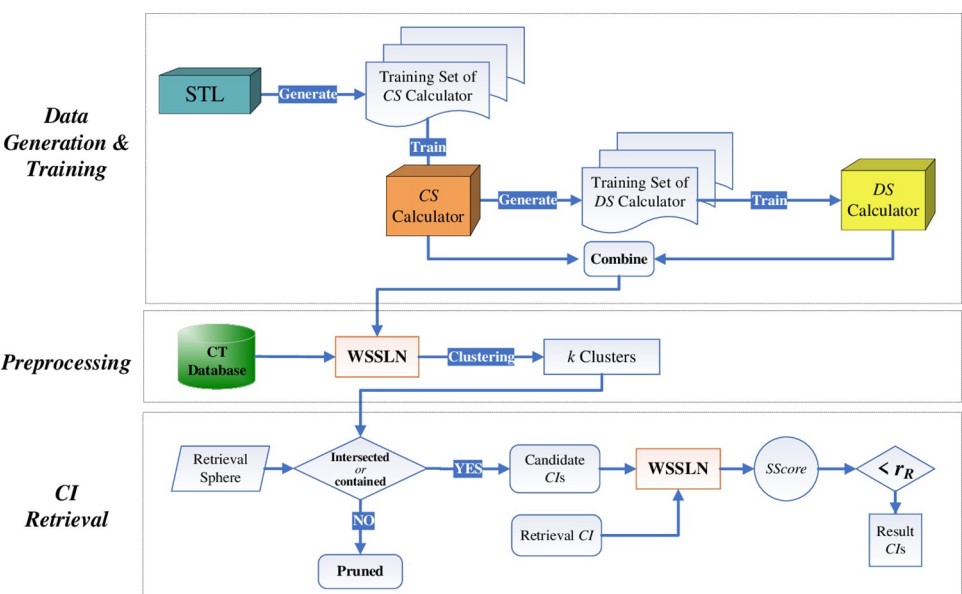

**Fig 1. The retrieval system framework.**

In the data generation and training stage, first use the STL to create the training set required for training the contour similarity(*CS*) calculator (see Section 3.2.3.2(A)). Then use the trained *CS* calculator to create training details similar to the training set required by the degree calculator (see Section 3.2.3.2(B)), and finally the *CS* calculator and the detail similarity(*DS*) calculator are merged to form our proposed *WSSLN*.

In the preprocessing stage, based on the *SScore* of each pair of *CI*s, the *CI*s are first grouped into *k* clusters by using AP-cluster algorithm [27]. Randomly select a *CI* as a cluster center in each cluster. Note that, the *SScore* represents the similarity of the outline and details of the two *CI*s based on *WSSLN* model, which is derived as:

$SScore(CI_i, CI_j) = \alpha \cdot Sim_C(CI_i, CI_j) + (1 - \alpha) \cdot Sim_D(CI_i, CI_j)$, where $Sim_C(CI_i, CI_j)$ (ref. Section 3.2.3.1)and $Sim_D(CI_i, CI_j)$ (ref. Section 3.2.3.2) refer to the contour similarity and detail similarity of the two *CI*s, respectively, the parameter $\alpha$ is set to be 0.5 by default.

In the *CI* retrieval stage, given a retrieval $CI(CI_R)$ and retrieval radius($r_R$), choose the clusters which are intersected with or contained by the retrieval sphere as the affected clusters. Calculate the *SScore* of $CI_R$ and the *CI*s ($CI_i$) falling in the affected clusters. If the *SScore* is less than or equal to $r_R$, then add $CI_i$ as the result *CI*s.

## 3.2 The WSSLN model

As one of the most important component in the retrieval system, in this section, we present an overall architecture of the *WSSLN* model for lung *CI* similarity measurement. As illustrated in Fig 2, the *WSSLN* model has two main modules: *the dataset generator* and *the similarity calculator*.

**3.2.1 Dataset generator.** As a key module to realize the weak supervised learning of the network, the dataset generator in Fig 3 is mainly used to generate a pair of lung *CI* pairs containing whether or not the labels are similar, which is used to generate a training set for subsequent training. It contains a STL, which is primarily employed to produce the training data for the *CS* calculator. In addition, the *CS* calculator that has successfully completed the training

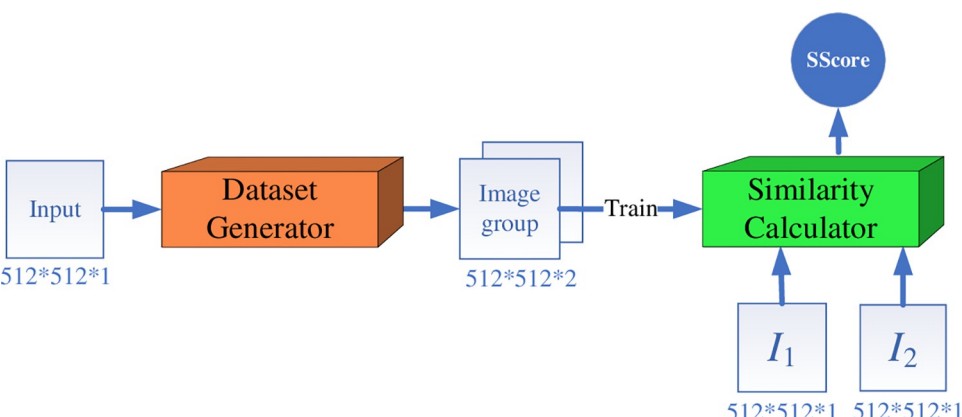

**Fig 2. The overall architecture of the *WSSLN*.**

provides the training set that is needed for further *CS* calculator training. Therefore, the *CS* calculator is not only a similarity calculator, but also a dataset generator.

Fig 4 shows the internal architecture of the STL. *U* is an input *CI* matrix (512*512*1), *V* is an output *CI* matrix (512*512*1), and each STL consists of a *grid generator(GG)* and a *sampler*. *θ* is a randomly generated 25*2 tensor, which enters *GG* after normalization. The purpose of the *GG* is to get the corresponding value of each pixel point of the output feature map. Then enter the sampler to insert the corresponding point into the new matrix *U* using the thin-slab sample interpolation transformation [22]. Finally get the output matrix *V* after spatial transformation.

**3.2.2 Similarity calculator.** In Fig 5, as a core part of the *WSSLN* for network training and similarity calculation, the similarity calculator includes the *Spatial Transformation Network*(STN) [23], the *CS calculator* and the *DS calculator*. Therefore, the lung *CI* similarity output by the *WSSLN* is the result of considering both contour similarity and detail similarity.

*3.2.2.1 The STN.* The STN is used to adjust the scale angle and other information of the two input lung *CIs* to make the position information of the two lung *CIs* consistent.

*3.2.2.2 The CS calculator.* In the lung *CI*, some lung diseases have a significant impact on the contour shape of the lung lobe, such as lobar pneumonia, atelectasis, etc. At the same time, the contour also represents the approximate position of the current lung *CI* in the whole lung to a certain extent. The *CS* calculator is responsible for the contour similarity calculation of the lungs in the two lung *CIs* without taking into account the details of the interior of the lungs.

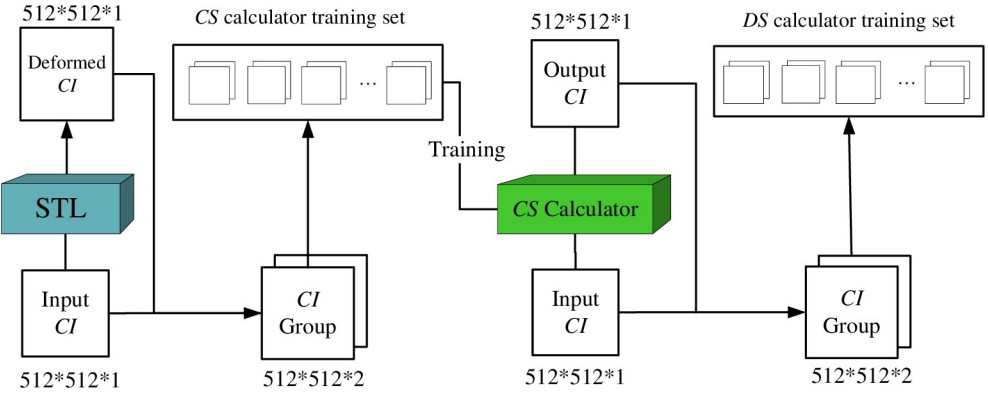

**Fig 3. The overall architecture of the data generator.**

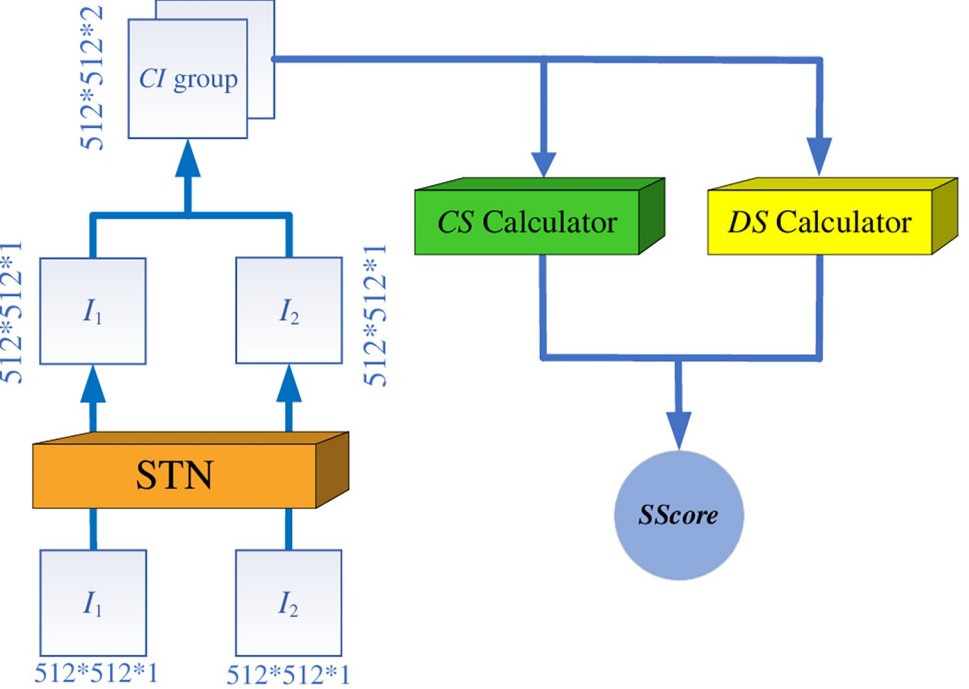

**Fig 4. The internal architecture of the STL.**

**Definition 1.** *Given two CIs: $CI_1$ and $CI_2$, their contour similarity($Sim_C$) can be expressed as*:

$$Sim_C = S(CI_1, CI_2) \qquad (1)$$

*where $S(CI_1,CI_2)$ denotes the contour similarity learning function.*

Based on Definition 1, the function $S(CI_1,CI_2)$ is implemented by a deep learning network that will be introduced below. The scale size of both $CI_1$ and $CI_2$ is 512*512*1.

**Fig 5. The architecture of the similarity calculator.**

Due to the continuity and diversity of contour shapes of lung *CI*s, there is a contextual linkage between different *CI* block(*CIB*)s of the two *CI*s. Given two *CI*s (e.g., $CI_1$ and $CI_2$), they are divided into several *CIB*s, respectively. Then we have $CI_1 = \{B_{11}, B_{12}, ..., B_{1k}\}$, $CI_2 = \{B_{21}, B_{22}, ..., B_{2k}\}$, where $k$ is the number of *CIB*s. The *CIB* $B_{1j}$ is tiled and expanded to form a vector $vec_{1j}$ that is synthesized into an *CIB* vector $vec_j = (vec_{1j}, vec_{2j})$ with the vector $vec_{2j}$, which is formed from the *CIB* $B_{2j}$.

**Definition 2.**   *Given k CIB vectors(i.e., $vec_1$, $vec_2$, ..., $vec_k$), their corresponding contextual linkage(CL) between CIB vectors can be derived in Eq.(2)*:

$$CL = \beta_1^T \cdot vec_1 + \beta_2^T \cdot vec_2 + ... + \beta_k^T \cdot vec_k \tag{2}$$

where CL is a constant, $\beta_i$ is the weight coefficient of the vector, and $vec_i$ is a k-dimensional column vector.

Based on the contextual linkage of different *CIB*s of the lung *CI*s in Definition 2, and some *CIB*s are important and others are secondary. The weight coefficients of some vectors may be significantly larger than those of other vectors. A self-attentive mechanism [24] is introduced in the *CS* calculator to filter out a small amount of important information in which the lung lobe contour part that changes significantly is given more weight. Therefore, a *Vision Transformer*(ViT) is used as a network architecture for the *CS* calculator, which pays more attention to the lung lobe contour part.

*3.2.2.3 The DS calculator.* Once a set of contour similar *CI*s are obtained through the contour similarity calculation, the *CI*s with similar information on bronchi, blood vessels, nodules, etc. inside the lung lobes of the input *CI* need to be further identified in terms of the detail similarity. The *DS* calculator is responsible for the detail similarity calculation of the *CI*s. The details refer to the soft tissues such as fine particles and blood vessels in the lung lobes. Some diseases will cause obvious changes in the internal details of the lung lobes, such as pulmonary nodules. As shown in Fig 6, the *DS* calculator focuses entirely on the parenchymal part of the lung which can be defined as *a pathological target area*(PTA) in the *CI*s.

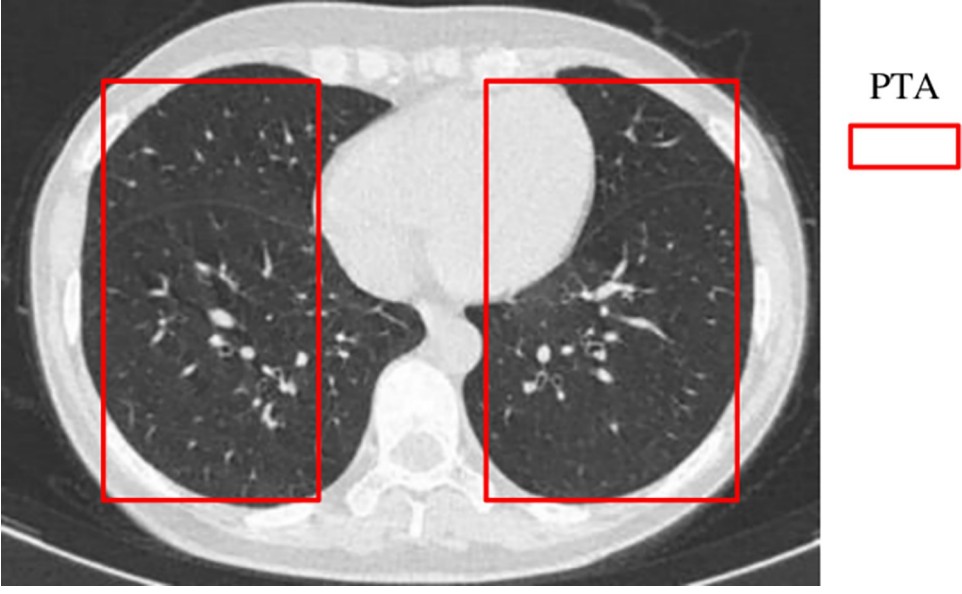

**Fig 6. Two *PTAs* in a lung *CI*.**

**Definition 3.** *Given a lung CI, its corresponding pathological target area (PTA) can be defined below*:

$$PTA = \{WS_1, WS_2, \ldots, WS_{|PTA|}\} \tag{3}$$

where $WS_i$ represents the i-th white stripe(WS) in the lung and |PTA| denotes the number of the WSs in the lung.

The *WS* in Definition 3 refers to the objects (e.g., *bronchi*, *blood vessels*, and *nodules*, etc) that are shown in the lung lobes of the *CIs*. Since the *WSs* have different shapes and sizes with their unique patterns in the *CIs*, it is hard to effectively discriminate them. So a graph model is adopted to describe the *WS*.

**Definition 4.** *Given a WS (i.e., $WS_i$), it can be represented by a graph model: $WS_i = \{V, E\}$, where V denotes the set of vertices of the graph, $V = \{v_1, v_2, \ldots, v_k\}$, $V_i$ denotes the i-th pixel point that is not equal to zero after binarization; E denotes the set of edges and $E = \{e_1, e_2, \ldots, e_n\}$, in which $e_k = <v_i, v_j>$ means that $v_i$ is connected to $v_j$.*

Based on Definition 4, two pixels are regarded adjacent if the Euclidean distance between them at their respective positions in the matrix does not exceed $\sqrt{2}$ (i.e., the distance between two adjacent pixels is 1). A vertex $v_i$ is taken as the center and a breadth search is performed around it to connect all adjacent vertices to generate the edge set *E*. Finally, a connected graph *G* is formed, which is the formation process of the *WS*.

Suppose that $PTA_1$ refers to the candidate lung parenchyma and $PTA_2$ means the lung parenchyma that has to be retrieved. The similarity of lung internal details in the two lung *CIs* can be described in Definition 5, which is determined by the number of vertices and the shape constructed by all the vertices.

**Definition 5.** *Given two PTAs(i.e., $PTA_1$ and $PTA_2$), whether they are similar can be defined by*:

$$bSimilar(PTA_1, PTA_2) = \begin{cases} TRUE, & if \sum_{i=1}^{|PTA_1|}\sum_{j=1}^{|PTA_2|} WS_{1i} \sim WS_{2j} \geq \theta \cdot |PTA_1| \\ FALSE, & otherwise \end{cases} \tag{4}$$

*where $WS_{ij}$ denotes the j-th stripe in the i-th PTA(i.e., $PTA_i$), |PTA| refers to the number of WSs in the PTA, and θ denotes the similarity threshold. If two WSs are similar, then $WS_{1i} \sim WS_{2j} = 1$, else $WS_{1i} \sim WS_{2j} = 0$.*

Fig 7 shows the internal structure of the *DS calculator*. According to Definition 5, the detail similarity of the lungs is highly dependent on the number, shape and position of the *WSs* inside the lobe. Since the fully connected layer at the Resnet18 [25] has some translation invariance, it can be replaced with a more position-sensitive convolutional layer as a variant of the Resnet18 that is used to construct the *DS* calculator.

The convolution operation is represented by the orange rectangle in Fig 7. The term '*SD*' means the quantity of convolution kernel moves. The term '*PD*' refers to the blank fill size around the *CI*. The batch normalization layer is denoted by '*BNL*'. The terms '*Relu*' and '*MPL*' stand for the maximum pooling layer and an activation function, respectively. Conv1 and four basicblocks make up the five blocks of which the entire *DS* calculator is composed. The fundamental residual principle of the Resnet is to perform an add operation on the basicblock input matrix and basicblock output matrix in the basicblock. When $CI_1$ and $CI_2$ are ready as the *DS* calculator inputs, after passing through the conv1 and four basicblocks, then the output of the *DS* calculator is '$Sim_D$' that indicates detail similarity of the two *CIs*.

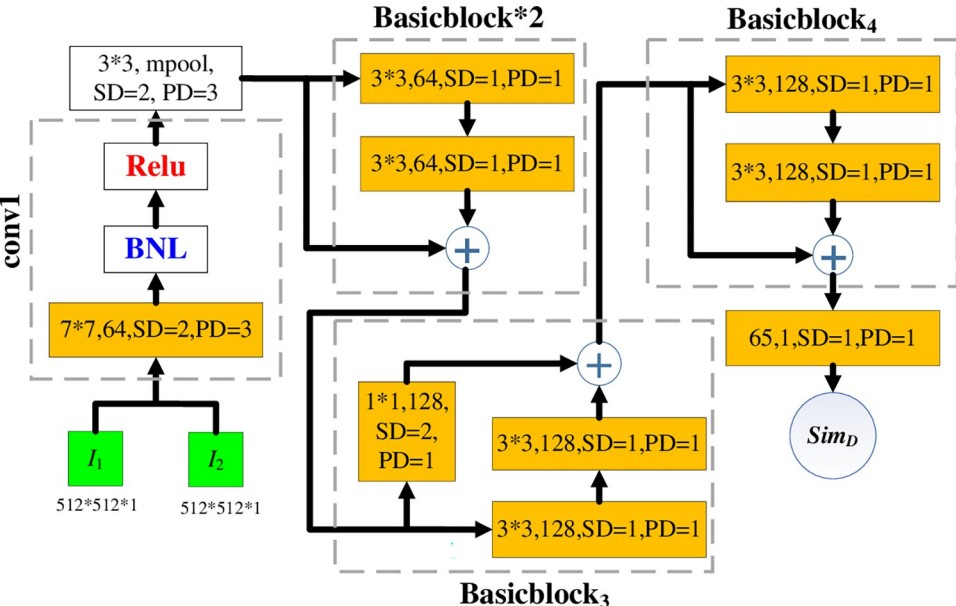

**Fig 7. The network architecture for the *DS* calculator.**

**3.2.3 Training.** *3.2.3.1 Data pre-processing.* In this paper, we use the LUNA16 (Lung Nodule Analysis 16) public dataset [26] that has a total of 1018 cases. For the *WSSLN* model, to effectively learn the features of the *PTA*, the lung parenchyma information from the lung *CI* needs to be extracted.

As a *CI* can be used to reflect the level of X-ray absorption by organs or tissues with gray-scale values, the density level of organ tissues in the *CI* quantitatively can be evaluated, which is known as CT value in HU (Hounsfield unit). Different CT values that correspond to various *CI* gray values are used to binarize the *CI*. Then the lung mask is obtained by dividing the external air and the internal torso using a seed filling algorithm. Since the lung contains numerous fibers, it appears to be hollow (relative to the lung). The closing operation in morphology fills these hollows. Fig 8 shows the changes before and after *CI* data preprocessing.

*3.2.3.2 Training set.* A. *CS calculator dataset.* First, the STL is used to generate the dataset and labels needed for the *CS* calculator. First, given a $CI_1$ (512*512*1) as an input, it can be

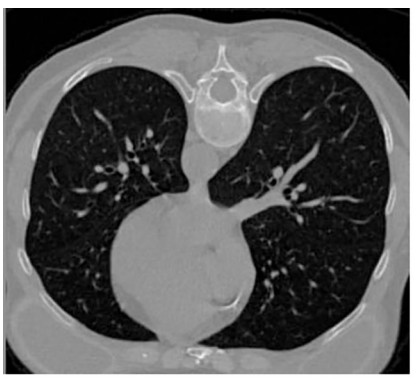

(a). Before

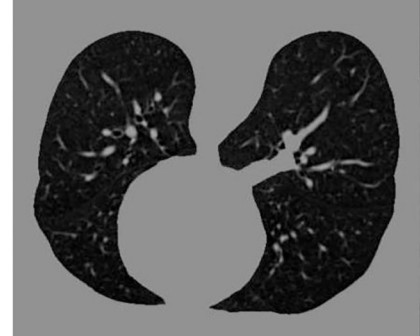

(b). After

**Fig 8. Comparison between before and after *CI* pre-processing.** (a). Before. (b). After.

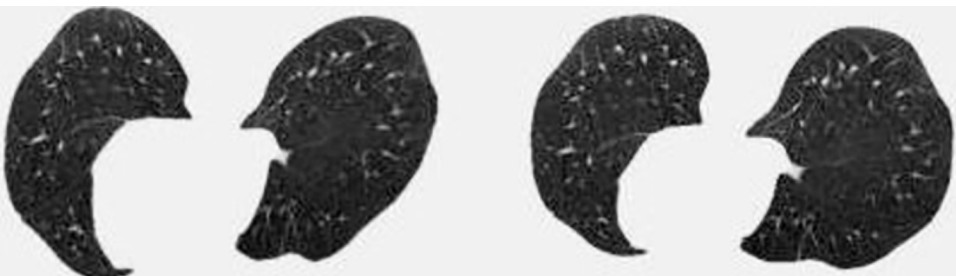

**Fig 9. Contour similar *CI* group.**

slightly deformed by the STL to generate $CI_2$ (512*512*1). In most cases, they are considered to be similar, which is synthesized into a 2*512*512 tensor and given label 1 (similar). However, a dataset with a label of 1 does not allow the *CS* calculator to learn which *CI*s are contour dissimilar to each other. So the next is to find the set of *CI*s with dissimilar contours. Since a CT scan case consists of hundreds of slices, the contour shape of different sections of the lung may change dramatically. *CI*s with contour dissimilarity to the input *CI* can be found in the *CI*s of the same case with different levels of sections. Synthesize them into a 2*512*512 tensor with the label 0. Figs 9 and 10 show the groups of the lung *CI*s with similar and dissimilar contours identified by the above method, respectively.

*B. DS calculator dataset.* After the *CS* calculator is trained, it can be used to randomly retrieve the database for *CI* $CI_2$ (512*512* 1) with contour similarity higher than a certain threshold to the input one $CI_1$ (512*512*1). Assume that these two *CI*s are detail dissimilar, then assign the label 0 (dissimilar) to this set of *CI*s and synthesize it into a tensor of 2*512*512. Next, it is necessary to find the *CI*s with similar details to be given to the *DS* calculator to learn. Given an input *CI* $CI_1$ (512*512*1), the $CI_2$ in the same group of *CI*s with adjacent layers must be similar to $CI_1$ in details, i.e., in a group of *CI*s, if $CI_1$ is the $r$-th layer lung section, then the $CI_2$ with similar details to it is the ($r$-1)-th layer section or the ($r$+1)-th layer section. In this way, a set of lung *CI*s with similar details can be found, given the label 1 (similar), and synthesized as a 2*512*512 tensor.

Since a *CI* with similar to $CI_1$ in details may be identified by the *CS* calculator, it is uncertain to retrieve a set of *CI*s with dissimilar details by our proposed method. The ratio of similar to dissimilar labels in the training set is adjusted to 3:1, allowing the *DS* calculator to learn more similar details. Figs 11 and 12 illustrate the groups of similar and dissimilar lung *CI*s identified by the above method.

*3.2.3.3 Loss function.* The network's objective in the lung *CI* similarity computation is to rate the similarity between two *CI*s. It is essentially transformed to a categorization task, i.e.,

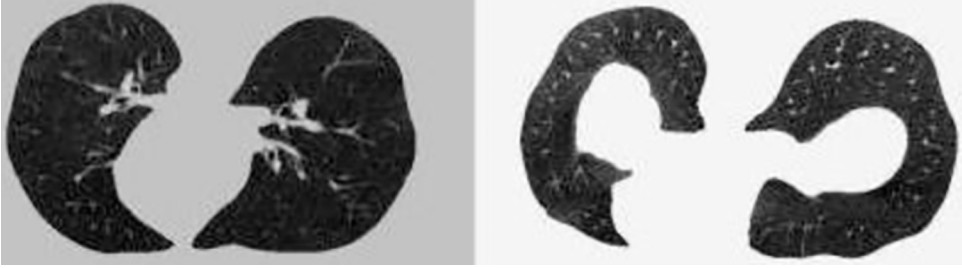

**Fig 10. Contour dissimilar *CI* group.**

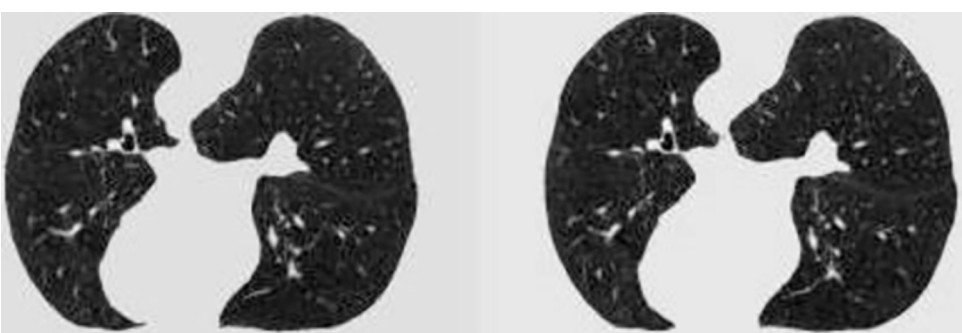

**Fig 11. Detail similar *CI* group.**

similar or not. For the training of the similarity calculator (i.e., *the CS calculator* and *the DS calculator*), the *sigmoid* function is adopted as the activation function and the cross-*entropy* function is adopted as the loss function. The difference between the predicted and the ground truth is measured using the loss function shown in Eq (5):

$$Loss(s(x), y) = \frac{1}{n} \sum_{x \in |batch|} \{-y \cdot \log[g(s(x))] - (1-y) \cdot log[1 - g(s(x))]\} \tag{5}$$

where $|batch|$ means the batch size in the network training, $s(x)$ refers to the similarity score of the output in the network and $s(x) \in [0,1]$. $y \in \{0,1\}$ denotes the corresponding ground truth, and $g(x)$ is the *sigmoid* function shown in Eq (6):

$$g(x) = \frac{1}{1 + e^{-x}} \tag{6}$$

## 3.3 Distance-based pruning scheme

The increase in the amount of *CI* data will lead to a significant decline in the performance of sequential retrieval based on the *WSSLN* model. To boost the retrieval performance, we propose a distance-based pruning scheme to effectively reduce the number of the similarity comparisons.

In the preprocessing step, as shown in Fig 13, the *n* lung *CI*s in the database are first grouped into *k* clusters by using the AP-clustering algorithm [27]. A (*n×n*) similarity matrix is calculated as an input of the clustering algorithm. For each cluster ($Cls_j$), we randomly choose a *CI* as a cluster center ($C_j$).

**Definition 6(*Retrieval Sphere*).** Given a retrieval CI($CI_R$) and a retrieval radius($r_R$), its corresponding retrieval sphere can be denoted as: $\Phi(CI_R, r_R)$.

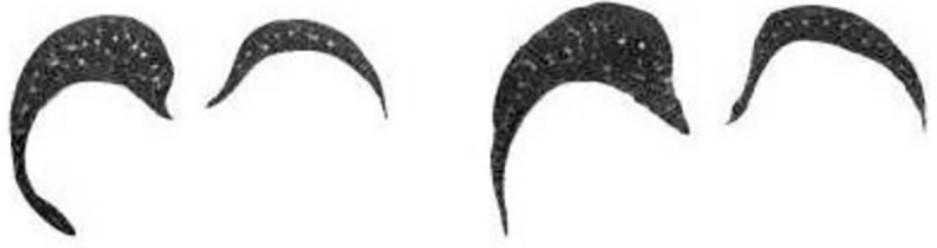

**Fig 12. Detail dissimilar *CI* group.**

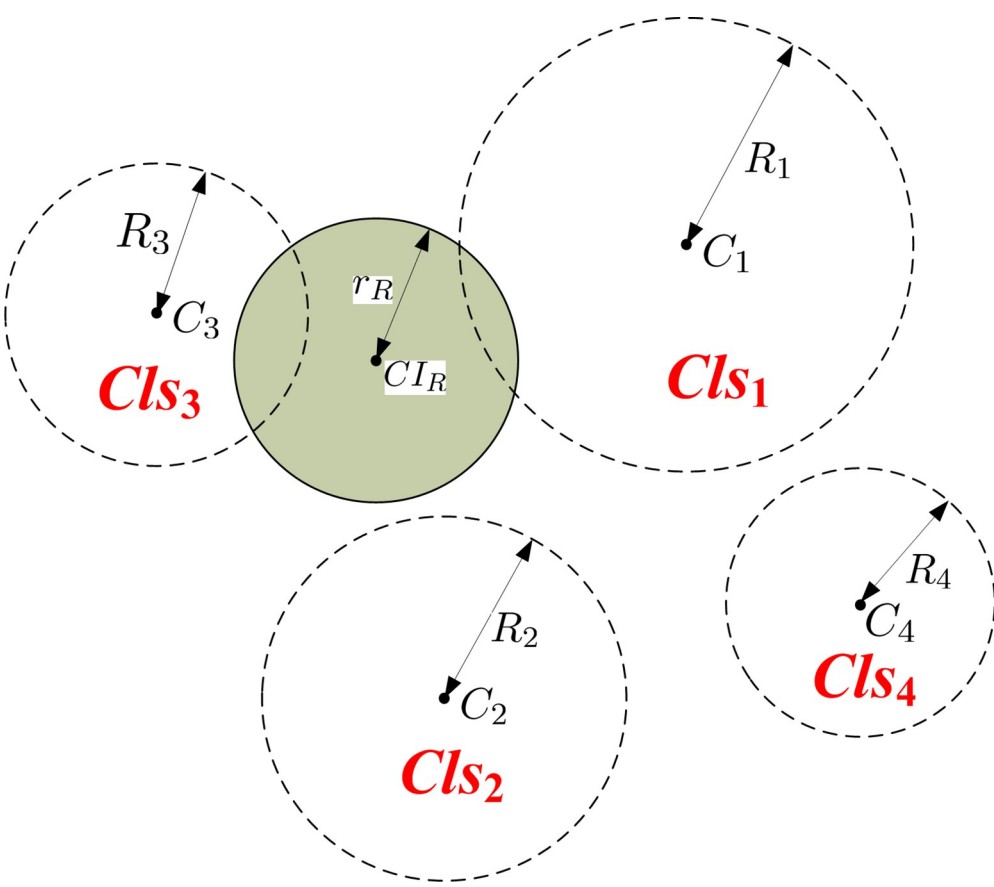

**Fig 13. A distance-based pruning example.**

**Definition 7(*Affected Cluster*, ACls).** Given a retrieval sphere(i.e., $\Phi(CI_R, r_R)$), a cluster ($Cls_j$) is an affected cluster if it is intersected with or contained by $\Phi(CI_R, r_R)$, formally represented as: $ACls = \{Cls_j | SScore(CI_R, \bar{C}_j) > R_j + r_R\}$, where$^-$refers to all cases where condition * is not met.

In Fig 13, for a retrieval sphere $\Phi(CI_R, r_R)$, its corresponding affected clusters is $Cls_1$ and $Cls_3$. The $CI$s falling in the $Cls_2$ and $Cls_4$ can be quickly filtered without high-cost similarity computation. Algorithm 1 summarizes the detailed steps of the pruning-based content-based $CI$ retrieval algorithm.

```
Algorithm 1. CTRetrieval(CI_R, r_R)
Input: CI_R: a retrieval CI, r_R: retrieval radius
Output: the result CIs
1. All Cls_j are regarded as ACls;
2. for each Cls_j do
3.    if SScore(C_j,CI_R)>R_j+r_R then remove Cls_j from the ACls;
4. end for
5. for each CI_i in each ACls_j do
6.    if SScore(CI_i,CI_R)<r_R then add CI_i as the result CIs;
7. end for
8. return the result CIs
```

## 4. Results and discussion

In this section, we conduct in-depth experiments to evaluate the effectiveness of the *WSSLN*. The PyTorch open-source deep-learning package is used to implement our proposed method, and an NVIDIA 1080Ti GPU is utilized for expedited training. The system runs on the following platform configuration information: Intel i5-11400F CPU, 16GB running memory, and 4T mechanical hard drive. The network is trained using the Adam optimizer with an initial learning rate of 0.001.

The *WSSLN*-based retrieval method is compared with three competitors, including two CNN-based hashing methods: the CNNSH [19] and the DSH [15]. One unsupervised method: the Locality Sensitive Hashing (LSH).

The database we used is the LUNA16 dataset [26] containing 44522 lung *CI*s. In this database, there are 179 sets of lung *CI*s. The number of each set containing the lung *CI*s of different levels ranges from 200 to 600, with an average number of 249 lung *CI*s per set.

### 4.1 A prototype system

The CBMIR prototype system is illustrated in Fig 14. The left side is the input window, which can be used to upload a retrieval *CI* and set the *k* value in Top-*k* retrieval by the Setting button. The right side window contains the output *k* similar *CI*s. A simple Top-*k* retrieval algorithm is designed in the system. The algorithm uses the *WSSLN* as the similarity learning function. After submitting a *CI* as the input, the similarity computation is conducted with all candidate *CI*s in the database and the similarities are sorted at the same time. Finally, the Top-*k* similar *CI*s can be obtained.

### 4.2 Effect of accuracy rate

In this experiment, the retrieval accuracy rate is represented by the *precision*.

$$Precision = \frac{TP}{TP + FP} \qquad (7)$$

where *TP* means the number of *CI*s output correctly and *FP* refers to the number of *CI*s output incorrectly.

Firstly, the *k* in Top-*k* retrieval is set to 5, which means retrieving the five lung *CI*s that are the most similar. Fig 15 illustrates a Top-*k* retrieval example, where the top row shows the

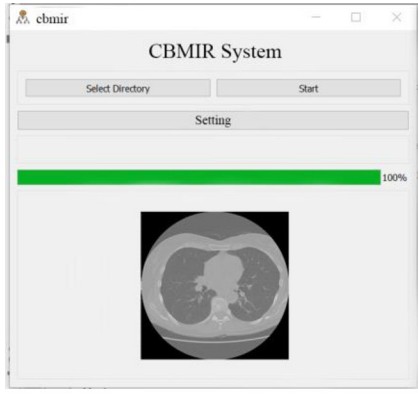 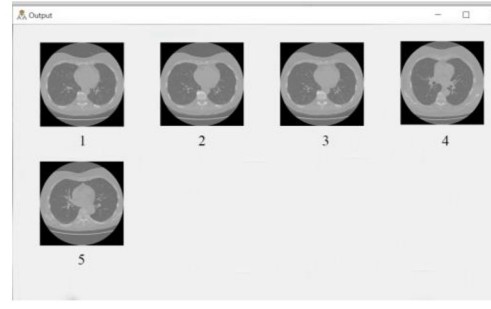

(a). Retrieval example                                    (b). Top-5 retrieval results

**Fig 14. An example of the prototype system.** (a). Retrieval example. (b). Top-5 retrieval results.

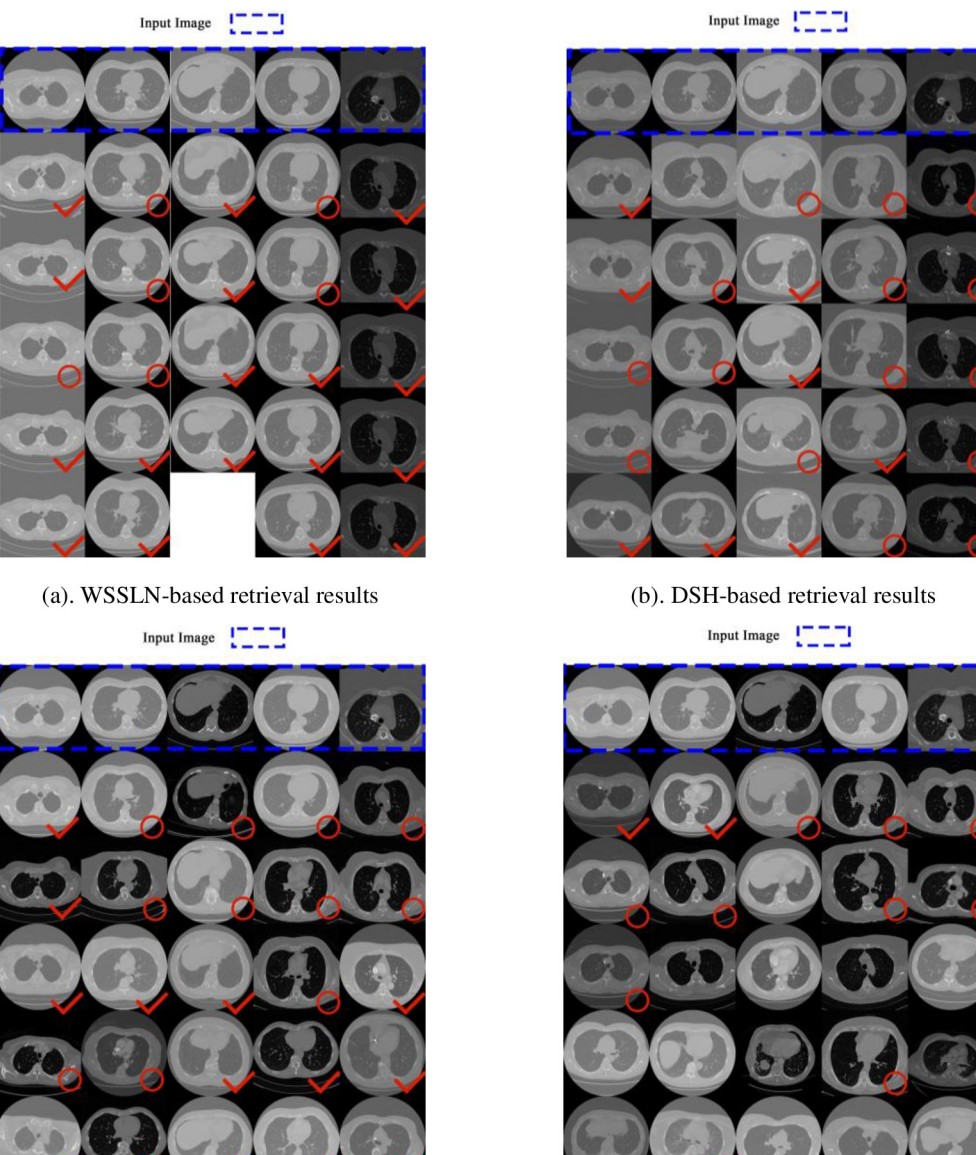

**Fig 15. Four examples of the top-5 retrievals.** (a). WSSLN-based retrieval results. (b). DSH-based retrieval results. (c). CNNSH-based retrieval results. (d). LSH-based retrieval results.

input lung *CI*s and the next five rows are the five *CI*s that are the most similar to the input one. The retrieval results based on the *WSSLN* approach are displayed in Fig 15A. Only four outputs are obtained by the system and are represented in the third column as four similar *CI*s. Fig 15B–15D represent the retrieval results of the DSH, the CNNSH, and the LSH, respectively. In the lower right corner of the output *CI*, a red tick is used to represent that it is similar to the input *CI* detail while the contour are similar, and a red circle indicates that it is similar to the input *CI* contour only. If both of the contour and detail similarities are considered, the accuracy of the *WSSLN*-based method is 72%, while the accuracies of the DSH-based method, the CNNSH-based method and the LSH-based method are 32%, 36% and 8%, respectively.

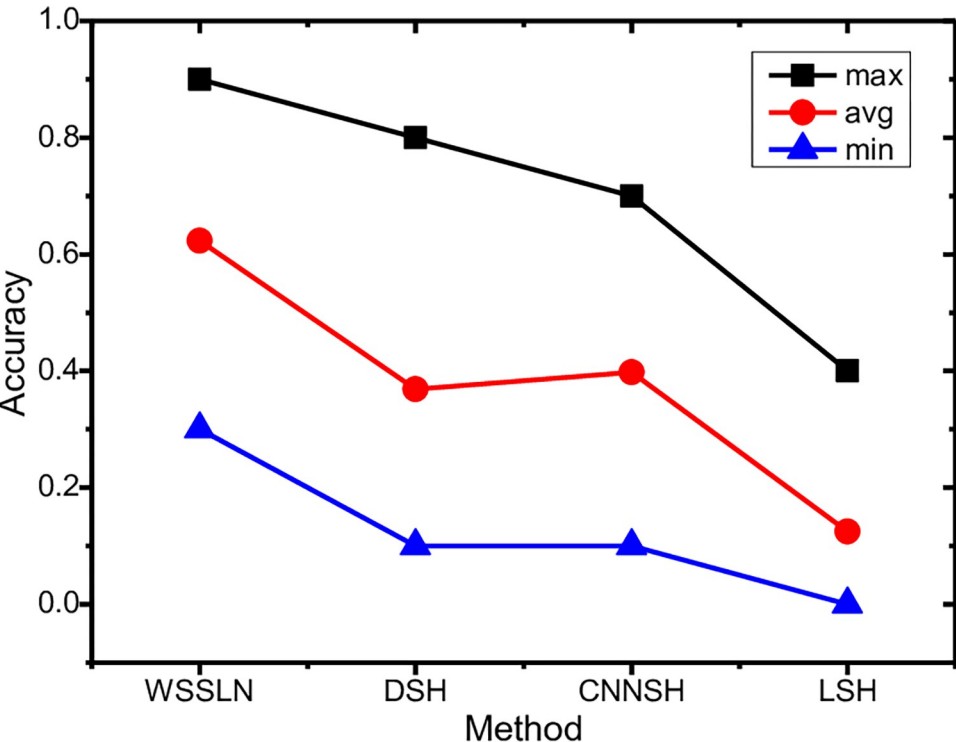

**Fig 16. Effect of accuracy rate.**

Therefore, for the similarity retrieval based on the contour and detail, the proposed *WSSLN* model achieves the highest precision compared to other three competitors. This is because the *WSSLN* model can better capture the intrinsic similarity of the lung *CI*s.

Secondly, the *k* of Top-*k* retrieval is set to 10, the retrieval accuracies of the four methods are compared in Fig 16 in which the *WSSLN* not only considers the similar contour but the similar details, and the rest of the methods are the same. Three metrics (i.e., *max*, *min* and *avg*) of the retrieval accuracy are provided. It's clear to see that the retrieval precision of the *WSSLN* is superior to the other three methods in all three metrics. The reason is the same to the above.

### 4.3 Comparison of mAP

Mean average precision(*mAP*) is also an important metric for evaluating a retrieval system, which is derived from the average precision(*AP*) defined in Eq (8):

$$AP = \frac{\sum Preci_{RK}}{Num_{RK}} \tag{8}$$

where *RK* means the index of the *CI* among all the *CI*s retrieved, $Preci_{RK}$ denotes the retrieval accuracy rate until the output *CI* with the index of *RK*, and $Num_{RK}$ refers to the total number of *RK*s. Then the formula of the *mAP* is derived below.

$$mAP = \frac{\sum_{i=0}^{n} AP_i}{n} \tag{9}$$

where *n* is the number of retrieval samples. According to Fig 17, if both of the contour and detail similarities are taken into account, our method's mAP@10 can reach 66.40%, compared

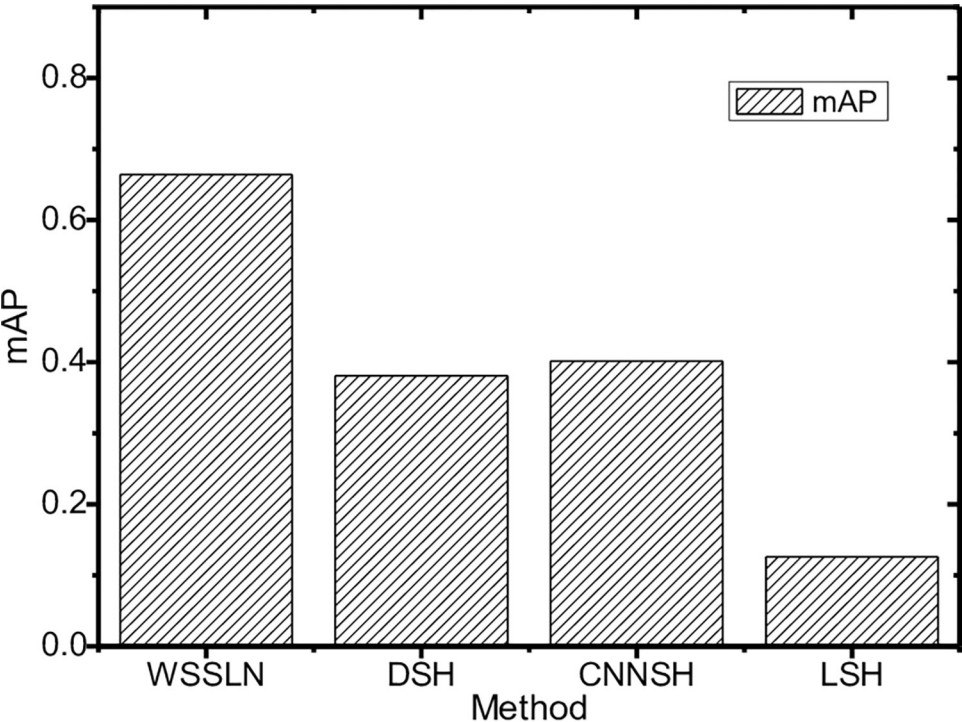

**Fig 17. Comparison of mAP.**

to the DSH's mAP@10 of 38.1%, the CNNSH's mAP@10 of 40.16%, and the LSH's mAP@10 of 12.56%. Both considering contour and detail similarities may make the retrieval system not able to find such similar lung *CI*s since they do not exist in the database, thus resulting in a relatively lower *AP*. So it makes sense that further analysis of the detail similarity would result in lower *mAP*s for all methods. According to the comparison of the four methods on *AP* and *mAP* in Fig 17, the *mAP* of the *WSSLN* method is significantly better than the other three ones. The reason is that the combination of two similar scales(i.e., the contour similarity and the detail similarity) can more effectively obtain similar *CI*s and improve retrieval accuracy.

## 4.4 Effect of the pruning scheme

The final experiment examines how the distance-based pruning scheme affects the retrieval performance. The number of lung *CI*s is 10000. We employ two techniques: 1) *pruning-based retrieval*; 2) *sequential scan*. Fig 18A and 18B show the effect of the pruning scheme in terms of the retrieval radius and data size, respectively. As illustrated in Fig 18A, the pruning-based method outperforms the sequential scan by a significant margin as the retrieval radius increases. In Fig 18B, the performance gap widens as the number of *CI*s rises because the pruning technique can effectively reduce the search space, improving the retrieval performance.

## 5. Conclusion

In this paper, we present an effective and efficient *WSSLN*-based CBSR method of the large lung *CI*s. Compared with the state-of-the-art deep learning-based CBMIR techniques, the main advantage of our proposed method is its weak supervision, i.e., the lack of a need to hire a professional physician to label the *CI*s for the network's training assignment. Moreover, under weakly supervised training, the *WSSLN*–based retrieval technique achieves satisfactory

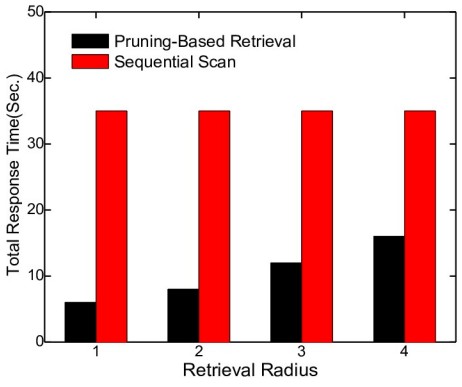

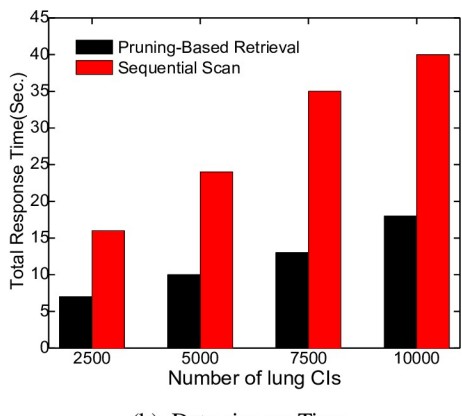

(a). Retrieval radius *vs*. Time    (b). Data size *vs*. Time

**Fig 18. Effect of the pruning scheme.** (a). Retrieval radius *vs*. Time. (b). Data size *vs*. Time.

performance. The extensive experiments on the real dataset indicated that our proposed *WSSLN*-based CBSR technique performs significantly better than the other three competitors in terms of the retrieval accuracy. Meanwhile, compared to the sequential scan, the *WSSLN*-based retrieval efficiency can be greatly increased with the help of the distance-based pruning scheme.

## Acknowledgments

## Ethics statement

This study does not involve ethical issues since the experimental data we used is an open source dataset which is available at: **https://luna16.grand-challenge.org/**.

## Author Contributions

**Methodology:** Nan Jiang.

**Writing – original draft:** Yi Zhuang.

**Writing – review & editing:** Nan Jiang.

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
