## [Decision Letter · Decision Letter 0]

27 Feb 2023

PONE-D-22-09269Effective and Efficient Content-Based Similarity Retrieval of Large Lung CT Image Database Based on WSSEN ModelPLOS ONE

Dear Dr. Zhuang,

Thank you for submitting your manuscript to PLOS ONE. After careful consideration, we feel that it has merit but does not fully meet PLOS ONE’s publication criteria as it currently stands. Therefore, we invite you to submit a revised version of the manuscript that addresses the points raised during the review process.

Please revise your paper according to the following suggested changes;

1. The authors need to re-write the abstract again to show the objective of this paper clearly.

2. Introduction section needs to be finished with summary and the contribution should be pointed out. 

3. The proposed approach each stage is not discussed in details, nor each stage is interlink appropriately and the discussion is very poor. Please revise it carefully.

4. The author also needs to make a clear conclusion/novelty at the end of each sub-section of methodology. 

5. In result section, the authors require to present clear details about the data analysis.

6. The following recent studies on should be discussed in related work;

Content-based image retrieval and semantic automatic image annotation based on the weighted average of triangular histograms using support vector machine, 2018

Query-by-visual-search: multimodal framework for content-based image retrieval, 2020

Boosting the performance of the BoVW model using SURF–CoHOG-based sparse features with relevance feedback for CBIRA novel image retrieval based on rectangular spatial histograms of visual words, 2018A Novel Image Retrieval Based on a Combination of Local and Global Histograms of Visual Words, 2016

We look forward to receiving your revised manuscript.

Kind regards,

Zahid Mehmood, PhD

Academic Editor

PLOS ONE

Journal Requirements:

“This work was supported in part by Zhejiang Provincial Natural Science Foundation of China under Grant No.Y22F021788; the Zhejiang Public Welfare Technology Application Research Project under grant No.GF22H185665; the Zhejiang Medical and Health Research Project under grant No. 2019RC070.”

“No competing interests”

Reviewers' comments:

Reviewer's Responses to Questions

**Comments to the Author**

1. Is the manuscript technically sound, and do the data support the conclusions?

Reviewer #1: Partly

Reviewer #2: Yes

2. Has the statistical analysis been performed appropriately and rigorously? 

Reviewer #1: N/A

Reviewer #2: Yes

3. Have the authors made all data underlying the findings in their manuscript fully available?

Reviewer #1: Yes

Reviewer #2: Yes

4. Is the manuscript presented in an intelligible fashion and written in standard English?

Reviewer #1: Yes

Reviewer #2: Yes

5. Review Comments to the Author

Reviewer #1: -The paper should be interesting ;;;

-please add block diagram of the proposed research;;;

-What is the result of the analysis?;;

-please add photos of the application of the proposed research, 2-3 photos ;;;

-figures should have high quality;;;

-what will society have from the paper?;;

-please try to compare the proposed analysis with other papers for example:

"Role of Hybrid Deep Neural Networks (HDNNs), Computed Tomography, and Chest X-rays for the Detection of COVID-19",

"A Novel Method for COVID-19 Diagnosis Using Artificial Intelligence in Chest X-ray Images".

-Conclusion: point out what have you done;;;;

-please add some sentences about future work;;;

-references should be from the web of science 2020-2022 (50% of all references, 30 references at least);;;

Reviewer #2: Author required to compare the proposed results with the existing method.

Parameters analysed for the results are very minimal for accurate finding need to incorporated more parameters

Complexity of the system was not discussed anywhere in the manuscript

6. PLOS authors have the option to publish the peer review history of their article (what does this mean?). If published, this will include your full peer review and any attached files.

Reviewer #1: No

Reviewer #2: No

---

## [Author Response · Author response to Decision Letter 0]

23 Mar 2023

Dear Editors & Reviewers,

First, thanks so much for your insightful and valuable comments which help us further improve the quality of this paper. Please review the following feedbacks. 

Q1. The authors need to re-write the abstract again to show the objective of this paper clearly.

A1: The abstract has been updated. The objective of the paper is study of retrieving similar CT image(CI)s to an input one from the large-scale CI database of labeled diseases which is helpful to realize precise computer-aided diagnosis.

Q2. Introduction section needs to be finished with summary and the contribution should be pointed out. 

A2: The summary and the contributions have been highlighted at the end of Section 1. 

Q3. The proposed approach each stage is not discussed in details, nor each stage is interlink appropriately and the discussion is very poor. Please revise it carefully.

A3: Each stage in our proposed approach has been revised and updated carefully.

Q4. The author also needs to make a clear conclusion/novelty at the end of each sub-section of methodology. 

A4: We have made a clear description of the contributions in each subsection of methodology.

Q5. In result section, the authors require to present clear details about the data analysis.

A5: The reasons and data analyses for almost all experiments have been updated in Section 4.

Q6. The following recent studies on should be discussed in related work;

Content-based image retrieval and semantic automatic image annotation based on the weighted average of triangular histograms using support vector machine, 2018

Query-by-visual-search: multimodal framework for content-based image retrieval, 2020

Boosting the performance of the BoVW model using SURF–CoHOG-based sparse features with relevance feedback for CBIR

A novel image retrieval based on rectangular spatial histograms of visual words, 2018

A Novel Image Retrieval Based on a Combination of Local and Global Histograms of Visual Words, 2016

A6: We have updated the related work.

---

## [Decision Letter · Decision Letter 1]

27 Apr 2023

Effective and Efficient Content-Based Similarity Retrieval of Large Lung CT Images Based on WSSLN Model

PONE-D-22-09269R1

Dear Dr. Zhuang,

We’re pleased to inform you that your manuscript has been judged scientifically suitable for publication and will be formally accepted for publication once it meets all outstanding technical requirements.

Kind regards,

Zahid Mehmood, PhD

Academic Editor

PLOS ONE

Additional Editor Comments (optional):

Reviewers' comments:

Reviewer's Responses to Questions

**Comments to the Author**

1. If the authors have adequately addressed your comments raised in a previous round of review and you feel that this manuscript is now acceptable for publication, you may indicate that here to bypass the “Comments to the Author” section, enter your conflict of interest statement in the “Confidential to Editor” section, and submit your "Accept" recommendation.

Reviewer #1: All comments have been addressed

Reviewer #2: All comments have been addressed

2. Is the manuscript technically sound, and do the data support the conclusions?

Reviewer #1: Yes

Reviewer #2: Yes

3. Has the statistical analysis been performed appropriately and rigorously? 

Reviewer #1: Yes

Reviewer #2: Yes

4. Have the authors made all data underlying the findings in their manuscript fully available?

Reviewer #1: No

Reviewer #2: Yes

5. Is the manuscript presented in an intelligible fashion and written in standard English?

Reviewer #1: Yes

Reviewer #2: Yes

6. Review Comments to the Author

Reviewer #1: -----------------------------------------------------------------------------------------------------------------------------------------------------------------------------------------------------------------------------------------------------------------------------------------------------------------------------------------------------------------------------------------------------------------------------------------------------

Reviewer #2: Author has addressed the queries and now it is in accepted form.

Existing and proposed data are tabulated in the result section

7. PLOS authors have the option to publish the peer review history of their article (what does this mean?). If published, this will include your full peer review and any attached files.

Reviewer #1: No

Reviewer #2: No

---

## [Editor Report · Acceptance letter]

2 May 2023

PONE-D-22-09269R1 

Effective and Efficient Content-Based Similarity Retrieval of Large Lung CT Images Based on WSSLN Model 

Dear Dr. Zhuang:

I'm pleased to inform you that your manuscript has been deemed suitable for publication in PLOS ONE. Congratulations! Your manuscript is now with our production department. 

Kind regards, 

on behalf of

Dr. Zahid Mehmood 

Academic Editor

PLOS ONE